# Contamination by Antibiotic-Resistant Bacteria in Selected Environments in Thailand

**DOI:** 10.3390/ijerph16193753

**Published:** 2019-10-05

**Authors:** Visanu Thamlikitkul, Surapee Tiengrim, Narisara Thamthaweechok, Preeyanuch Buranapakdee, Wilai Chiemchaisri

**Affiliations:** 1Division of Infectious Diseases and Tropical Medicine, Department of Medicine, Faculty of Medicine Siriraj Hospital, Mahidol University, Bangkok 10700, Thailand; narisara.tha@mahidol.ac.th; 2Department of Clinical Microbiology and Applied Technology, Faculty of Medical Technology, Mahidol University, Nakhon Pathom 73170, Thailand; surapee.tie@mahidol.ac.th; 3Bureau of Environmental Health, Department of Health, Ministry of Public Health, Nonthaburi 11000, Thailand; nuchpakdee@gmail.com; 4Department of Environmental Engineering, Faculty of Engineering, Kasetsart University, Bangkok 10900, Thailand; fengwlc@ku.ac.th

**Keywords:** antimicrobial resistance, extended-spectrum beta-lactamase-producing Enterobacteriaceae, environment, Thailand

## Abstract

This study determined the presence of important antibiotic-resistant bacteria in selected environments in Thailand, including wastewater samples from 60 hospitals; washed fluid, leachate, flies, cockroaches, and rats collected from five open markets; washed fluid from garbage trucks; and stabilized leachate from a landfill facility. At least one type of antibiotic-resistant bacteria was isolated from all samples of influent fluid before treatment in hospitals, from wastewater treatment tank content in hospitals, and from 15% of effluent fluid samples after treatment with chlorine prior to draining it into a public water source. Antibiotic-resistant bacteria were recovered from 80% of washed market fluid samples, 60% of market leachate samples, all fly samples, 80% of cockroach samples, and all samples of intestinal content of rats collected from the open markets. Antibiotic-resistant bacteria were recovered from all samples from the landfill. Extended-spectrum beta-lactamase (ESBL)-producing *Escherichia coli* and/or *Klebsiella pneumoniae* were the most common antibiotic-resistant bacteria recovered from all types of samples, followed by carbapenem-resistant *E. coli* and/or *K. pneumoniae.* Colistin-resistant Enterobacteriaceae, carbapenem-resistant *Psuedomonas aeruginosa*, carbapenem-resistant *Acinetobacter baumannii*, colistin-resistant Enterobacteriaceae, vancomycin-resistant *Enterococci*, and methicillin-resistant *S. aureus* were less common. These findings suggest extensive contamination by antibiotic-resistant bacteria in hospital and community environment in Thailand.

## 1. Introduction

Antimicrobial resistance (AMR) has enormous adverse impact on humans in terms of morbidity, mortality, and economic loss [1]. The health and economic burden of AMR on humans in Thailand is enormous [2,3,4,5]. The World Health Organization (WHO), the World Organization for Animal Health (OIE), and the Food and Agriculture Organization of the United Nations (FAO) endorsed a global action plan on AMR in 2015 because AMR will affect sectors beyond human health, such as animal health, agriculture, food security, the environment, and economic development [6]. Therefore, the One Health concept, which is defined as a collaborative, multisectoral, and transdisciplinary approach at the local, regional, national, and global levels with the goal of achieving optimal health outcomes while recognizing the interconnection among people, animals, plants, and their shared environment, should be adopted to facilitate the prevention and containment of AMR. The global action plan on AMR emphasizes particularly important gaps in knowledge that need to be filled, including information on the magnitude of AMR in humans, animals, agriculture, and the environment; and, enhanced understanding regarding how resistance develops and spreads, including how resistance circulates within and between humans and animals, and through food, water, and the environment. Such information and understanding are important for the development of new tools, policies, and regulations to counter AMR.

The Thailand Antimicrobial Resistance Containment and Prevention Program was implemented as a One Health approach in 2012 [7]. One of the actions of the program is to determine AMR drivers and the dynamics of the AMR chain to improve our understanding of how AMR develops and spreads among humans, animals, agriculture, and the environment in Thailand. Several reports of the program regarding the magnitude of antibiotic-resistant bacteria in hospitalized patients, community, animals, and foods revealed that extended-spectrum beta-lactamase (ESBL)-producing Enterobacteriaceae, carbapenem-resistant Enterobacteriaceae, carbapenem-resistant *Acinetobacter baumannii*, carbapenem-resistant *Pseudomonas aeruginosa*, colistin-resistant Enterobacteriaceae, methicillin-resistant *Staphylococcus aureus*, and vancomycin-resistant Enterococci were common or important antibiotic-resistant bacteria in Thailand [8,9,10,11,12,13]. However, the information on antibiotic-resistant bacteria in the environment in Thailand is limited. Contamination by antibiotic-resistant bacteria in the Chao Phraya River and its tributaries [14], in flies [15,16], and in migratory birds [17] in Thailand was reported.

The environment has been increasingly recognized for the role it might play in the global spread of clinically relevant antibiotic resistance [18]. One of the proposed critical knowledge gaps and research needs related to the environmental dimensions of AMR is the contributions of different sources of antibiotic resistant bacteria in the environment [19]. Antibiotic-resistant bacteria from humans and animals always directly or indirectly contaminate the environment, which results in the environment becoming a reservoir for antibiotic-resistant bacteria that can be transmitted to humans and animals. Moreover, the environment also contains antibiotic residues from humans, animals, and foods that can induce resistance in the bacteria residing in the environment, and such the antibiotic-resistant bacteria can be transmitted to humans and animals. Therefore, the action plans to counter AMR has to establish and effectively implement legislation and/or regulations to prevent contamination of the environment with antimicrobials and antibiotic-resistant bacteria. Such legislation and/or regulations are usually deficient partly because the science needed to develop a policy is lacking, and this needs to be addressed. In 2017, the WHO asked its Member States to respond to a questionnaire on the status of in-country legislation and/or regulations to prevent contamination of the environment with antimicrobials and antibiotic-resistant bacteria. The responsible institute in Thailand responded that Thailand currently has no legislation regarding control of wastewater discharges containing antimicrobials and antibiotic-resistant bacteria into the environment because the information that is needed to draft and establish this type of legislation and/or regulations is presently insufficient.

Accordingly, the aim of this study was to determine the magnitude of antibiotic-resistant bacteria contamination in selected environments in Thailand, including wastewater samples from hospitals; from washed fluid, leachate, flies, cockroaches, and rats collected from open markets, from washed fluid from a garbage truck that collected garbage from the five open markets; and, from stabilized leachate from a landfill facility.

## 2. Materials and Methods

### 2.1. Study Sites and Collection of Samples During January and August 2018

#### 2.1.1. Study Hospitals

Sixty public hospitals were selected from a total of 959 public hospitals in Thailand by stratified random sampling to cover all geographical regions of Thailand, all sizes of hospitals, and the various types of hospital wastewater management systems. The study hospitals included 30 community hospitals, 20 general hospitals, and 10 regional hospitals. The wastewater management systems of the study hospitals included oxidation ditch (27 hospitals), conventional activated sludge (17 hospitals), sequential batch reactor (5 hospitals), aerated lagoon (4 hospitals), submerged aeration fixed film (4 hospitals), construction wetland (1 hospital), stabilization pond (1 hospital), and anaerobic filter-submerged fixed-film aeration tank (1 hospital). A total of 186 samples collected from the wastewater management system of each hospital included 5 liters of influent fluid before treatment (60 samples), 5 liters of content in wastewater treatment tank (66 samples), and 5 liters of effluent fluid after treatment with chlorine prior to draining it into a public water source (60 samples). All samples were collected in sterile closed containers and kept at 4 °C until they were sent to a microbiological laboratory within 24 h after collection.

#### 2.1.2. Open Fresh Markets

Five open fresh markets in Nonthaburi Province, which is located near Bangkok, were selected.

The samples collected from each market included 5 liters of washed market fluid (5 samples) and 5 liters of market leachate (5 samples). All samples were collected in sterile closed containers and kept at 4 °C until they were sent to microbiological laboratory within 24 h after collection.

At least five flies per market (5 samples) and at least 5 cockroaches per market (5 samples) were collected in separate plastic bags. All samples were sent to microbiological laboratory within 24 h after collection.

At least 2 rats caught from each market (5 samples) were sacrificed. The intestinal contents of the rats caught from the same market were collected with a swab in Cary-Blair transport medium, and the swabs were sent to microbiological laboratory within 24 h after collection.

#### 2.1.3. Landfill Facility

A landfill facility located in Nonthaburi Province, where the garbage from the 5 study open markets were disposed, was chosen. The samples collected from this landfill facility included 5 liters of washed fluid from a garbage truck that collected garbage from the markets in Nonthaburi Province (1 sample), and 5 liters of stabilized leachate from the landfill facility (1 sample). All samples were collected in sterile closed containers and kept at 4 °C until they were sent to microbiological laboratory within 24 h after collection.

### 2.2. Microbiological Procedures

Cultures of all samples were performed at the microbiology laboratory of the Division of Infectious Diseases and Tropical Medicine, Department of Medicine, Faculty of Medicine Siriraj Hospital, Mahidol University, Bangkok, Thailand. The targeted antibiotic-resistant bacteria to be isolated from the samples were extended-spectrum beta-lactamase (ESBL)-producing Enterobacteriaceae, carbapenem-resistant Enterobacteriaceae (CRE), carbapenem-resistant *A. baumannii*, carbapenem-resistant *P. aeruginosa*, colistin-resistant Enterobacteriaceae, vancomycin-resistant *Enterococci* (VRE), and methicillin-resistant *S. aureus* (MRSA).

#### 2.2.1. Microbiological Evaluation of Influent Fluid Samples and Content in Wastewater Treatment 

Tank Collected from Hospitals, Washed Market Fluid, Market Leachate, Washed Fluid from Garbage Truck, and Stabilized Leachate from Landfill Facility 

An undiluted sample of 0.1 mL was inoculated on MacConkey agar supplemented with ceftriaxone 2 µg/mL, on Enterococcosel agar (BBL, Becton Dickinson, USA) supplemented with vancomycin 6 µg/mL, and on mannitol salt agar supplemented with oxacillin 6 µg/mL. The inoculated agar plates were incubated at 35 °C for 18–24 h. 

A diluted sample with buffered peptone water supplemented with ceftriaxone 2 µg/mL at 1:10 was prepared. A diluted sample of 0.1 mL was also inoculated on all of the aforementioned agars and they were incubated at 35 °C for 18–24 h. 

Colony count of grown bacteria was performed on the plate inoculated with either undiluted or diluted sample. Each type of grown bacteria was inoculated on a blood agar plate and incubated at 35 °C for 18–24 h. The bacteria grown on blood agar plate were identified up to species level and were used for antibiotic susceptibility testing as described in Section 2.2.4.

#### 2.2.2. Microbiological Evaluation of the Effluent Samples Collected from Hospitals 

An undiluted sample of 1 mL was inoculated on MacConkey agar supplemented with ceftriaxone 2 µg/mL, on Enterococcosel agar (BBL, Becton Dickinson, USA) supplemented with vancomycin 6 µg/mL, and on mannitol salt agar supplemented with oxacillin 6 µg/mL. The inoculated agar plates were incubated at 35 °C for 18–24 h. 

A diluted sample with buffered peptone water supplemented with ceftriaxone 2 µg/mL at 1:10 was prepared. A diluted sample of 0.1 mL was also inoculated on all of the aforementioned agars and they were incubated at 35 °C for 18–24 h. 

Colony count of grown bacteria was performed on the plate inoculated with either undiluted or diluted sample. Each type of grown bacteria was inoculated on blood agar plate and incubated at 35 °C for 18–24 h. The bacteria grown on a blood agar plate were identified up to species level and were used for antibiotic susceptibility testing as described in Section 2.2.4.

#### 2.2.3. Microbiological Evaluation of Flies, Cockroaches, and Intestinal Swabs of Rats Collected from Markets

Tryptic Soy Broth (TSB) of 100 mL was added to each fly and cockroach sample and the mixture was blended in a food mixer device for one minute. The mixture was incubated at 35 °C for 18–24 h. Then, 0.1 mL of the incubated mixture was inoculated on MacConkey agar supplemented with ceftriaxone 2 µg/mL, on Enterococcosel agar (BBL, Becton Dickinson, USA) supplemented with vancomycin 6 µg/mL, and on mannitol salt agar supplemented with oxacillin 6 µg/mL. The swab containing intestinal content of rat was inoculated on the aforementioned agars and they were incubated at 35 °C for 18–24 h. Identification of the isolated bacteria and antibiotic susceptibility testing were performed as described in Section 2.2.4.

#### 2.2.4. Identification of Isolated Bacteria and Antibiotic Susceptibility Testing

Identification of isolated bacteria was performed by MALDI-TOF (BRUKER) [20]. Antibiotic susceptibility test of the isolated bacteria was performed by disk diffusion method using Mueller–Hinton agar according to the Clinical and Laboratory Standards Institute 2018 [21]. Enterobacteriaceae isolate was tested with ceftriaxone, ertapenem, and colistin. Production of ESBL in Enterobacteriaceae was performed by double-disk synergy tests using a coamoxiclav disk, ceftriaxone disk, and ceftazidime disk. *Acinetobacter baumannii* or *P. aeruginosa* isolates were tested with imipenem, meropenem, colistin. *Staphylococcus aureus* isolate was tested with cefoxitin. *Enterococcal* isolate was tested with vancomycin.

## 3. Results

### 3.1. Samples Collected from Wastewater Management System of Hospitals

#### 3.1.1. Prevalence of Antibiotic-Resistant Bacteria

The prevalence of antibiotic-resistant bacteria isolated from influent fluid before treatment (INF), content in wastewater treatment tank (CWW), and effluent fluid after treatment with chlorine prior to draining it into a public water source (EFF) from 188 samples collected from 60 hospitals are shown in Table 1. At least one type of antibiotic-resistant bacteria was isolated from all samples of INF and CWW collected from all hospitals, and from 15% of EFF samples. Among the nine hospitals that contained antibiotic-resistant bacteria in effluent fluid samples after treatment with chlorine, four of them used an oxidation ditch system, two used a sequential batch reactor system, and each of the other three hospitals used conventional activated sludge, submerged aeration fixed film, and stabilization pond system, respectively.

#### 3.1.2. Number of Isolated Bacteria from Samples

The amount of bacteria contained in the influent fluid before treatment (INF), in the content of the wastewater treatment tank (CWW), and in the effluent fluid after treatment with chlorine prior to draining it into a public water source (EFF) from 188 samples collected from 60 hospitals are shown in Table 2. The influent fluid before treatment, and the content in the wastewater treatment tank contained an abundant number of bacteria, whereas the number of bacteria was much decreased in the effluent fluid after treatment with chlorine. Most samples of effluent fluid after treatment with chlorine contained no bacteria, whereas some samples of effluent fluid after treatment with chlorine still contained bacteria.

#### 3.1.3. Types of Isolated Bacteria from Samples

The types of antibiotic-resistant bacteria isolated from the influent fluid before treatment (INF), from the content in the wastewater treatment tank (CWW), and from the effluent fluid after treatment with chlorine prior to draining it into a public water source (EFF) from 186 samples collected from 60 hospitals are shown in Table 3. The ESBL-producing *Escherichia coli* and/or *K. pneumoniae*, and carbapenem-resistant *E. coli* and/or *K. pneumoniae* were commonly isolated from the influent fluid before treatment and from the content in the wastewater treatment tank, but the prevalence of such antibiotic-resistant bacteria was much lower in the effluent fluid after treatment with chlorine. Carbapenem-resistant *P. aeruginosa* and carbapenem-resistant *A. baumannii* were less frequently isolated from the influent fluid before treatment, from the content in the wastewater treatment tank, and even further decreased in the effluent fluid after treatment with chlorine. Colistin-resistant Enterobacteriaceae and VRE were isolated from some samples of the influent fluid before treatment and from the content in the wastewater treatment tank; however, they were absent in the effluent fluid after treatment with chlorine. The MRSA was recovered from only one sample of the content in the wastewater treatment tank. Among the effluent fluid samples after treatment with chlorine with a presence of antibiotic-resistant bacteria collected from 9 hospitals (Table 1), seven samples contained ESBL-producing *E. coli* and/or *K. pneumoniae*, five samples contained carbapenem-resistant *E. coli* and/or *K. pneumoniae*, one sample contained carbapenem-resistant *P. aeruginosa*, and one sample contained carbapenem-resistant *A. baumannii*.

### 3.2. Samples Collected from Markets

#### 3.2.1. Washed Market Fluid and Market Leachate

The number of isolated bacteria and the prevalence and type of antibiotic-resistant bacteria isolated from the samples collected from markets are shown in Table 4. All samples contained a large number of bacteria. The washed market fluid samples from four markets (80%) contained ESBL-producing *E. coli*, whereas such samples from two markets (40%) contained carbapenem-resistant *E. coli* and/or *K. pneumoniae*. No colistin-resistant Enterobacteriaceae, MRSA, or VRE was recovered from any of these samples. The market leachate samples from three markets (60%) contained ESBL-producing *E. coli*, whereas none of these samples contained carbapenem-resistant *E. coli* and/or *K. pneumoniae*, colistin-resistant Enterobacteriaceae, MRSA, or VRE.

#### 3.2.2. Flies, Cockroaches, and Rats

All fly samples contained ESBL-producing *E. coli*. No carbapenem- or colistin-resistant Enterobacteriaceae, MRSA, or VRE was recovered from any of these samples. Cockroach samples from four markets (80%) contained ESBL-producing *E. coli*. No carbapenem- or colistin-resistant Enterobacteriaceae, MRSA, or VRE was recovered from any of these samples. All samples of intestinal content of rats contained ESBL-producing *E. coli*, and one such sample (20%) contained colistin-resistant *E. coli*. No carbapenem-resistant Enterobacteriaceae, MRSA, or VRE was recovered from any of these samples.

### 3.3. Samples Collected from Landfill Facility

The number of isolated bacteria, and the prevalence and type of antibiotic-resistant bacteria isolated from samples collected from the landfill facility are shown in Table 5. All samples contained a large number of bacteria. All samples contained ESBL-producing *E. coli*, whereas none of these samples contained carbapenem-resistant *E. coli* and/or *K. pneumoniae*, carbapenem- or colistin-resistant Enterobacteriaceae, MRSA, or VRE.

## 4. Discussion

The study sites that were selected for this study included both healthcare facility and community. Healthcare facilities were selected to include hospitals of various sizes that were located in all geographical regions of Thailand and that have different wastewater management systems so that these hospitals would be accurate representatives of public hospitals in Thailand, and so that their study results could be generalized to all public hospitals in Thailand. The study sites in community were open fresh markets and a landfill facility located in the same area as the markets in a province located near Bangkok. Therefore, they should be representatives of urban community in Thailand, and their study results could be generalized to other urban areas of Thailand. The targeted antibiotic-resistant bacteria included those causing infection and colonization of hospitalized patients, people, and food animals in community. The ESBL-producing Enterobacteriaceae were found to be the most common and important resistant bacteria in healthcare settings and in community. Carbapenem-resistant bacteria, MRSA, and VRE, usually cause infection and colonization in hospitalized patient. Colistin-resistant Enterobacteriaceae could cause infection and colonization in hospitalized patients who received colistin, and in food animals on farms that used colistin for prevention, control, and treatment of infection in food animal. The sample collection, sample transportation, and microbiological procedures for isolation and identification of the aforementioned targeted antibiotic-resistant bacteria in this study were appropriate, and an attempt was made to isolate all types of targeted antibiotic-resistant bacteria from the collected samples. Therefore, culturing of samples that should have heavy contamination with many types of bacteria, such as influent fluid sample from hospital before treatment, was performed using both undiluted and diluted samples, whereas the samples that were most likely to contain a minimal amount of bacteria, such as effluent fluid sample from hospital after chlorine treatment, were cultured using a larger amount of sample in order to minimize the risk of a false-negative culture result.

The abundant amount of bacteria and the presence of all kinds of common antibiotic-resistant bacteria causing infection in hospitalized patients in all samples of influent fluid before treatment and in the content of wastewater treatment tank observed in this study was expected because these samples were waste fluid drained from all areas of the hospitals, including the sinks and toilets from hospital wards. However, the samples of effluent fluid after treatment with chlorine prior to draining it into a public water source collected from nine hospitals (15% of all study hospitals) still contained antibiotic-resistant bacteria. This observation should not be related to the type of wastewater treatment system of such hospitals because these hospitals used five different wastewater treatment systems. Therefore, these positive finding should be associated with operational defects and shortcomings of the wastewater treatment system at these hospitals. The presence of antibiotic-resistant bacteria in effluent fluid from hospital is important since these antibiotic-resistant bacteria will contaminate the environment when they are drained into an open public water source, and hospital effluent was reported to be a reservoir for carbapenemase-producing bacteria [22]. Moreover, if contaminated effluent fluid is used for agriculture, the food products would be contaminated with antibiotic-resistant bacteria that can be transmitted to humans and animals. The hospitals with antibiotic-resistant bacteria in their effluent fluid were notified of the study results so that they could fix the problem. The responsible department of the Ministry of Public Health is considering implementing a regulation that hospital effluent fluid after treatment with chlorine must be periodically tested to ensure the absence of antibiotic-resistant bacteria prior to draining it into a public water source. However, the hospital effluent fluid without antibiotic-resistant bacteria draining into an open public water source still poses a threat from the presence of a large number of antibiotic-resistant genes of killed bacteria that might be horizontally transferred to other live bacteria residing in the environment [23]. Therefore, the use of hospital effluent fluid for consumption in humans and animals, and for agriculture should be avoided. Another important threat of hospital effluent fluid is that it contains antibiotic residues. A study of the magnitude and impact of antibiotic residues in the environment in hospital is currently being conducted.

The finding that only 60% to 80% of washed market fluid samples and market leachate samples contained antibiotic-resistant bacteria instead of all samples may be explained by the fact that detergents or other antiseptic agents were used for cleaning the markets, and these agents could kill bacteria contaminating these markets. It is clear from the findings of this study that the environment in open fresh markets and the landfill facility were important reservoirs of antibiotic-resistant bacteria in community. Flies, cockroaches, and rats caught from the study markets were reservoirs of antibiotic-resistant bacteria and were also vectors for transmission of antibiotic-resistant bacteria in the community. Flies were postulated to be one of the important vectors for transmission of antibiotic-resistant bacteria in the community, including on animal farms [24]. Although several studies from China and Thailand found colistin-resistant bacteria in flies [15,16,25], our study isolated only ESBL-producing *E. coli* from flies. This may be due to the fact that the flies in our study were caught in an urban area, whereas other studies were conducted in rural areas with animal farms where colistin-resistant bacteria were usually present. The ESBL-producing *E. coli* was also a predominant antibiotic-resistant bacterium isolated from cockroaches and rats in this study. Cockroaches and rats were also reported to be reservoirs for antibiotic-resistant bacteria in several countries [26,27,28,29]. Another important threat from the environment in community is that it would also contain antibiotic residues. A study on the magnitude and impact of antibiotic residues in the environment in community is also currently being conducted. 

The findings from this study strongly suggest that there is an extensive amount of contamination by antibiotic-resistant bacteria in the environment in both hospital and communities in Thailand. Establishment of legislation and regulation for effective wastewater treatment system in hospital to avoid a presence of antibiotic-resistant bacteria in the treated effluent draining into public water sources is being considered based on the results of this study. However, it is much more difficult to minimize contamination by antibiotic-resistant bacteria in the environment that is caused by flies, cockroaches, and rats in community. Everyone has to comply with personal health sanitation and hygiene in order to avoid acquisition of antibiotic-resistant bacteria from the environment, and to prevent transmission of antibiotic-resistant bacteria to the environment while awaiting legislation and/or regulations to prevent contamination of the environment in community with antimicrobial-resistant bacteria. 

## 5. Conclusions

Antibiotic-resistant bacteria, especially ESBL-producing *E. coli* and/or *K. pneumoniae*, were recovered from wastewater samples from hospitals even after treatment; washed fluid, leachate, flies, cockroaches, and rats collected from open markets; washed fluid from garbage truck; and stabilized leachate from a landfill facility in the community. These findings strongly suggest that there is an extensive amount of contamination by antibiotic-resistant bacteria in the environment in both hospital and communities in Thailand.

## Figures and Tables

**Table 1 ijerph-16-03753-t001:** Type and number of samples, and number of samples with isolated antibiotic-resistant bacteria recovered from samples collected from hospitals.

Type of Sample	Number of Samples	Number of Samples with Antibiotic-Resistant Bacteria
INF	60	60 (100%)
CWW	66 *	66 (100%)
EFF	60	9 (15.0%)

* Some hospitals had more than one sample. Abbreviations: INF, influent fluid before treatment; CWW, content in wastewater treatment tank; EFF, effluent fluid after treatment with chlorine prior to draining it into a public water source.

**Table 2 ijerph-16-03753-t002:** The number of isolated bacteria from samples collected from hospitals.

Type of Sample	Number of Samples	Number of Isolated Bacteria (CFU/mL)	
		Range	Median
INF	60	<100–>10^5^	2.5 × 10^4^
CWW	66 *	600–>10^5^	2.5 × 10^4^
EFF	60	0–>10^5^	0

* Some hospitals had more than one sample. Abbreviations: CFU, colony-forming unit; INF, influent fluid before treatment; CWW, content in wastewater treatment tank; EFF, effluent fluid after treatment with chlorine prior to draining it into a public water source.

**Table 3 ijerph-16-03753-t003:** Antibiotic-resistant bacteria isolated from the samples collected from hospitals.

	Number of Samples		
	INF	CWW	EFF
Total samples	60	66	60
Samples with antibiotic-resistant bacteria			
ESBL-producing *E. coli*	12 (20.0%)	13 (19.7%)	0
ESBL-producing *K. pneumoniae*	7 (11.7%)	9 (13.6%)	1 (1.7%)
ESBL-producing *E. coli* and ESBL-producing *K. pneumoniae*	34 (56.7%)	40 (60.6%)	6 (10.0%)
Carbapenem-resistant-*E. coli*	4 (6.7%)	6 (9.0%)	2 (3.3%)
Carbapenem-resistant- *K. pneumoniae*	27 (45.0%)	28 (42.4%)	1 (1.7%)
Carbapenem-resistant-*E. coli* and *K. pneumoniae*	5 (8.3%)	6 (9.0%)	2 (3.3%)
Carbapenem-resistant-*P. aeruginosa*	2 (3.3%)	4 (6.1%)	1 (1.7%)
Carbapenem-resistant-*A. baumannii*	4 (6.7%)	1 (1.5%)	1 (1.7%)
Colistin-resistant Enterobacteriaceae	5 (8.3%)	8 (12.1%)	0
VRE	5 (8.3%)	12 (18.2%)	0
MRSA	0	1 (1.5%)	0

Abbreviations: INF, influent fluid before treatment; CWW, content in wastewater treatment tank; EFF, effluent fluid after treatment with chlorine prior to draining it into a public water source; ESBL, extended-spectrum beta-lactamase; *E. coli*, *Escherichia coli*; *K. pneumoniae*, *Klebsiella pneumoniae*; *P. aeruginosa*, *Pseudomonas aeruginosa*; *A. baumannii, Acinetobacter baumannii*; VRE, vancomycin-resistant Enterococci; MRSA, methicillin-resistant *Staphylococcus aureus*.

**Table 4 ijerph-16-03753-t004:** The number of isolated bacteria, prevalence, and type of antibiotic-resistant bacteria isolated from the samples collected from markets.

Type of Sample	Number of Sample	Number of Sample with Antibiotic-Resistant Bacteria	Number of Isolated Bacteria (CFU/mL)	
			Range	Median
Washed market fluid sample	5	4 (80%) -ESBL-producing *E. coli* (4) -Carbapenem-resistant *E. coli* (1) -Carbapenem-resistant *K. pneumoniae* (1)	6 × 10^3^–1 × 10^5^	1.7 × 10^4^
Market leachate sample	5	3 (60%) -ESBL-producing *E. coli* (3)	5 × 10^3^–>10^5^	7.4 × 10^4^

Abbreviations: CFU, colony-forming unit; ESBL, extended-spectrum beta-lactamase; *E. coli*, *Escherichia coli*; *K. pneumoniae*, *Klebsiella pneumoniae*.

**Table 5 ijerph-16-03753-t005:** Number of isolated bacteria, prevalence, and type of antibiotic-resistant bacteria isolated from samples collected from a landfill facility.

Type of Sample	Number of Samples	Number of Samples with Antibiotic-Resistant Bacteria	Number of Isolated Bacteria (CFU/mL)	
			Amount	Median
Washed fluid from garbage truck	1	1 (100%) -ESBL-producing *E. coli*	5 × 10^5^	NA
Stabilized leachate in landfill facility	1	1 (100%) -ESBL-producing *E. coli*	1.3 × 10^5^	NA

Abbreviations: CFU, colony-forming unit; ESBL, extended-spectrum beta-lactamase; *E. coli*, *Escherichia coli*; N/A, not available.

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
