# Peer review of "Contamination by Antibiotic-Resistant Bacteria in Selected Environments in Thailand"

_ijerph, 2019, doi:10.3390/ijerph16193753_

Round 1

Reviewer 1 Report

This paper reported the determination of the magnitude of antibiotic-resistant bacteria contamination in selected environments in Thailand. The study sites that were selected for this study included both healthcare facility and community. Antibiotic-resistant bacteria, especially ESBL producing E. coli and/or K. pneumoniae, were recovered from wastewater samples from hospitals even after treatment; washed fluid, leachate, flies, cockroaches, and rats collected from open markets; washed fluid from garbage truck; and, stabilized leachate from a landfill facility in community. My opinion is that this article after minor revisions can be published in International Journal of Environmental Research and Public Health:

- The implementation of the microbiological measurements seem to be correct but it also should be mentioned the corresponding standardized methods suitable for the microbial investigation of these samples.

- The authors stated that the samples from hospitals contained antibiotic-resistant bacteria even after the waste water treatment procedures. Is there any data to compare the efficiency the different waste water treatment methods? Which is the most effective method?

- The paper suggest extensive contamination by antibiotic-resistant bacteria in environment in both hospital and community in Thailand. Is it possible to compare the results of these study data with international data?

Author Response

Reviewer 1

Comment

- The implementation of the microbiological measurements seems to be correct but it also should be mentioned the corresponding standardized methods suitable for the microbial investigation of these samples.

Response

We referred to the standard methods for microbial investigation as said in reference 20 and 21.

Comment

- The authors stated that the samples from hospitals contained antibiotic-resistant bacteria even after the waste water treatment procedures. Is there any data to compare the efficiency the different waste water treatment methods? Which is the most effective method?

Response

The method section described all 5 types of waste water treatment system of all hospitals and the number of the hospitals that used particular waste water treatment system. The result section described the types of waste water treatment system of 9 hospitals that still contained antibiotic-resistant bacteria after the waste water treatment procedures. The waste water treatment systems of these 9 hospitals were not confined to particular type of waste water treatment system. Therefore, the discussion section said ‘This observation should not be related to the type of waste water treatment system of such hospitals because these hospitals used five different waste water treatment systems. Therefore, these positive finding should be associated with operational defects and shortcomings of the waste water treatment system at these hospitals.’

Comment

- The paper suggests extensive contamination by antibiotic-resistant bacteria in environment in both hospital and community in Thailand. Is it possible to compare the results of these study data with international data?

Response

We found no studies on contamination by antibiotic-resistant bacteria in both hospital and community environment in the same study. We referred to a study (reference 22) reporting that hospital effluent was a reservoir for carbapenemase-producing bacteria.

Reviewer 2 Report

The manuscript reports important findings which highlight the situation on presence of antibiotic-resistant bacteria in selected environments in Thailand. The manuscript is concise and should be accepted for publication after some revision.

Most of my remarks are editorial remarks. Firstly, in several instances in the text, articles ("the"; "a") are missing, e.g. line 20: determined the presence; e.g. line 110, to a microbiological laboratory

Abstract

Lines 36-37, in hospital and community environments

Introduction

Line 75, contaminate

Materials and methods

Line 105, and several more times in the text/tables: which figure is correct, 188 or 186? Or were 188 samples taken and 2 of them not analysed (186 results were reported)?

Lines 145 and 157, diluted samples … were prepared

Lines 149, 161, 171: please refer to 2.2.4

Results

Line 231, from Table 3, I count six carbapenem-resistance E.coli and/or K. pneumoniae sample, the test mentions four, please correct accordingly

Line 253, Table 4, please add in third column /K. pneumoniae after ESBL-producing E. coli

Discussion

Line 280, accurate representatives

Line 287, were found

Line 323, if the waste water plant treated effluents are not void of such bacteria and/or remaining DNA which can pose a risk to human health, what measures can the authors think of to avoid reintroduction in the environment leading to contamination of soils used for agriculture?

Lien 327, I think that 5 samples is a too low number for reliable conclusions and statistics, but I agree that it gives a hint about the situation

Line 348, what is the supposed meaning of "legislation and regulations are being considered"? Are the respective pieces of legislation currently updated taking into account the latest knowledge?

Author Response

Reviewer 2

Comment

Most of my remarks are editorial remarks. Firstly, in several instances in the text, articles ("the"; "a") are missing, e.g. line 20: determined the presence; e.g. line 110, to a microbiological laboratory

Response

They are corrected according to the reviewer’s suggestion.

Comment

Abstract

Lines 36-37, in hospital and community environments

Response

Corrected as suggested.

Comment

Introduction

Line 75, contaminate

Response

Corrected as suggested.

Comment

Materials and methods

Line 105, and several more times in the text/tables: which figure is correct, 188 or 186? Or were 188 samples taken and 2 of them not analysed (186 results were reported)?

Response

‘188 samples’ was corrected to ‘186 samples’ for all places.

Comment

Lines 145 and 157, diluted samples … were prepared

Response

Corrected to 186 samples as suggested.

Comment

Lines 149, 161, 171: please refer to 2.2.4

Response

Corrected as suggested.

Comment

Results

Line 231, from Table 3, I count six carbapenem-resistance E.coli and/or K. pneumoniae sample, the test mentions four, please correct accordingly

Response

The data in the table 3 are correct. We corrected the texts that described the table 3 in order to correspond with the data in table 3.

Comment

Line 253, Table 4, please add in third column /K. pneumoniae after ESBL-producing E. coli

Response

The data in the table 4 are correct. We corrected the texts that described the table 4 in order to correspond with the data in table 4.

Comment

Discussion

Line 280, accurate representatives

Response

Corrected as suggested.

Comment

Line 287, were found

Response

Corrected as suggested.

Comment

Line 323, if the waste water plant treated effluents are not void of such bacteria and/or remaining DNA which can pose a risk to human health, what measures can the authors think of to avoid reintroduction in the environment leading to contamination of soils used for agriculture?

Response

Antibiotic-resistant bacteria can be killed by waste water treatment system which is appropriately operated and no viable antibiotic-resistant bacteria should be present in the treated effluent as mentioned in the discussion section. However, DNA of killed bacteria is still present in the treated effluent but a transmission of DNA from killed antibiotic-resistant bacteria to other bacteria in the environment is not simple and this effect should have less importance than a presence of viable antibiotic-resistant bacteria in the treated effluent draining into public water sources or being used for agriculture. Therefore, the measure to avoid reintroduction in the environment leading to contamination of antibiotic-resistant bacteria in soils used for agriculture is efficient waste water treatment system whereas the measure to eliminate DNA of killed bacteria from waste water is not available.

Comment

Line 327, I think that 5 samples is a too low number for reliable conclusions and statistics, but I agree that it gives a hint about the situation.

Response

We agreed. However, the allocated budget allowed us to collect the samples from only 5 markets.

Comment

Line 348, what is the supposed meaning of "legislation and regulations are being considered"? Are the respective pieces of legislation currently updated taking into account the latest knowledge?

Response

This sentence is changed to ‘Establishment of legislation and regulation for effective waste water treatment system in hospital to avoid a presence of antibiotic-resistant bacteria in the treated effluent draining into public water sources is being considered based on the results of this study.’
